# How Do Synchronous Lung Metastases Influence the Surgical Management of Children with Hepatoblastoma? An Update and Systematic Review of the Literature

**DOI:** 10.3390/cancers11111693

**Published:** 2019-10-31

**Authors:** Roberta Angelico, Chiara Grimaldi, Carlo Gazia, Maria Cristina Saffioti, Tommaso Maria Manzia, Aurora Castellano, Marco Spada

**Affiliations:** 1Division of Abdominal Transplantation and Hepatobiliopancreatic Surgery, Bambino Gesù Children’s Hospital IRCCS, 00165 Rome, Italy; roberta.angelico@gmail.com (R.A.); chiara.grimaldi@opbg.net (C.G.); mcristina.saffioti@opbg.net (M.C.S.); 2Department of Surgery Science, HPB and Transplantation Unit, University of Rome Tor Vergata, 00133 Rome, Italy; carlogazia9@gmail.com (C.G.); tomanzia@libero.it (T.M.M.); 3Division of Oncohematology, Bambino Gesù Children’s Hospital IRCCS, 00165 Rome, Italy; aurora.castellano@opbg.net

**Keywords:** lung metastases in hepatoblastoma, high-risk hepatoblastoma, liver resection, liver transplantation, metastasectomy

## Abstract

Approximately 20% of children with hepatoblastoma (HB) have metastatic disease at diagnosis, most frequently in the lungs. In children with HB, lung metastatic disease is associated with poorer prognosis. Its treatment has been approached with a variety of methods that integrate chemotherapy and surgical resection. The timing and feasibility of complete extirpation of lung metastases, by chemotherapy and/or metastasectomy, is crucial for the surgical treatment of the primary liver tumor, which can vary from major hepatic resections to liver transplantation (LT). In children with unresectable HB, which can be surgically treated only by LT, the persistence of unresectable metastases after neoadjuvant chemotherapy excludes the possibility of recurring to LT with consequent negative impact on patients’ outcomes. Due to limited evidence and experience, there is no consensus amongst oncologists and surgeons across institutions regarding the surgical treatment for HB with synchronous metastatic lung disease. This narrative review aimed to update the current management of pulmonary metastasis in children with HB and to define its role in the decision-making strategy for the surgical approach to primary liver tumours.

## 1. Introduction

Hepatoblastoma (HB) is the most common paediatric primary liver tumour, generally detected within the third year of life [1], with an increasing annual incidence of 1.5 cases per million [2]. According to different protocols, the overall survival in HB has dramatically increased in the last years mainly owing to advances in chemotherapy (CHT), reaching an overall 5-year survival of 70–80% [3]. The current strategy of HB treatment is based on different approaches including delayed tumor resection after pre-operative chemotherapy (CHT) or, in selected cases, upfront resection with or without post-operative CHT.

HB can metastasize in several organs, including the adrenal gland, colon, diaphragm, kidney, and lungs. It is estimated that around one fifth of children with HB present pulmonary metastasis at diagnosis [4]. This phenomenon is one of the poorest prognostic factors in HB. In the presence of lung metastasis, the 5-year event-free survival (EFS) is approximately 21–28%, while the overall survival ranges from 25 to 50% [5,6].

Lung metastasis are generally defined as a single nodule more than 10 mm or several nodules with at least one more than 5 mm. Computed tomography (CT) is the gold standard in the diagnosis of lung metastasis, and it is usually performed as a routine exam during the diagnostic assessment [7]. However, this technique still faces some limitations in paediatric solid tumours despite the remarkable technologic improvements. Although the CT scan is highly sensitive, it lacks specificity for differential diagnosis between malignant and benign lung nodules. Furthermore, its resolution is not always able to detect the smallest nodules [7].

In the last decades, cisplatin-based CHT has achieved excellent results in children with HB [1,4,8,9]. Cisplatin is either used alone or combined with doxorubicin, 5–fluorouracil and/or vincristine for advanced HB. Although the optimal schedule and doses for a chemotherapeutic regimen are not yet defined, worldwide several HB therapeutic protocols have been adopted and all international studies demonstrated that CHT is of the utmost importance to treat HB with synchronous lung metastases. Indeed, CHT achieves up to 80% metastasis resolution [6]. When lung metastasis are not resolved by CHT or relapse occurs, surgical metastasectomy is an effective alternative option [6].

The timing and feasibility of complete extirpation of lung metastasis, by medical and/or surgical treatment, is crucial for the surgical approach to the primary liver tumor, which can vary from major hepatic resections to liver transplantation (LT). In children with unresectable HB, which can be surgically treated only by LT, the persistence of unresectable metastases after neoadjuvant chemotherapy exclude the possibility of recurring to LT with consequent negative impact on patients’ outcomes. Due to limited evidence and experience, there is no consensus amongst oncologists and surgeons across institutions regarding the surgical treatment for HB with synchronous metastatic lung disease. Thus, the present study aimed to review and describe the outcomes of HB with synchronous lung metastasis in consideration of possible surgical approaches, for both primary and secondary tumours, to the best of the available literature.

## 2. Material and Methods

### 2.1. Search Strategy

A systematic search was performed to identify relevant studies focused on the therapeutic management and outcomes of children affected by HB with lung metastases at diagnosis. The search strategy complied with the Preferred Reporting Items for Systemic Reviews and Meta-Analysis (PRISMA) guidelines [10]. A search of the electronic databases MEDLINE-PubMed, Ovid Medline, and EMBASE was conducted using the following search terms: hepatoblastoma, liver transplantation, liver resection, metastasis, high-risk, advanced. Studies published before 21 May 2019 were considered.

### 2.2. Screening Process

The present qualitative systematic review included a priori search criteria of journal articles amongst children (<18 years) affected by HB. Records of children affected by HB with lung metastasis present at diagnosis that detailed the outcomes of the pulmonary metastasis and the surgical management of the primary liver tumour were included. Studies were limited to the English language.

Exclusion criteria were studies that lacked surgical details, did not report outcomes of lung metastasis, and described HB metastasis other than pulmonary. Additionally, review articles, non-clinical studies, guidelines, case reports, expert opinions, letters to the editor, and conference summaries were excluded. All studies that originated from the same centre were considered and analyzed, and the possible overlap of clinical cases reported across the studies was evaluated.

### 2.3. Study Selection and Data Extraction

A total of 564 articles were observed. Two reviewers (RA and CG) independently screened the identified studies and their data was extracted. In case of disagreement, the paper was discussed by all the authors. After the systematic screening, 56 studies were identified for systematic review analysis (Figure 1). Given the paucity of patients identified within the selection criteria in each selected study, the results are reported as a narrative review.

Data extracted from each study included (where available): year of publication, inclusion criteria, exclusion criteria, number of participants, and the data-relevant outcome variables explained above.

## 3. Hepatoblastoma (HB) Risk Stratification

In the last two decades, HB risk stratification has been studied by several international groups who actively cooperated and contributed to improve outcomes in patients affected by HB: the Children’s Oncology Group (COG), the International Childhood Liver Tumour Strategy Group (SIOPEL), the Children’s Cancer Group (CCG) and the Pediatric Oncology Group (POG), the German Society for Paediatric Oncology and Haematology (GPOH) and the Japanese Study Group for Pediatric Liver Tumors (JPLT).

Worldwide, the pre-treatment extent of disease (PRETEXT) system is the most-used staging tool for HB. It determines the liver sectors affected by HB before any treatment commences [11]. Each patient diagnosed with HB is assigned to a PRETEXT group (I, II, III, or IV) based on the number of contiguous uninvolved sections of the liver. Moreover, annotation factors are included in the PRETEXT staging system, defining the involvement of major vascular structures of the liver and extrahepatic sites. Annotation factors comprise “V”: involvement of the retrohepatic inferior vena cava or involvement of all three major hepatic veins; “P”: involvement of the portal vein; “E”: extrahepatic involvement of contiguous structures; “M”: distant metastatic disease; “F”: multifocal liver tumours; “R”: tumour rupture [12,13]. Multiple trials have confirmed that the PRETEXT stage is a powerful predictor of patients’ survival, and drives the medical and surgical treatment of HB. Therefore, the PRETEXT and the annotation factors have been adopted in the current Pediatric Hepatic International Malignancy Therapeutic Trial (PHITT), which is the largest ongoing single clinical trial undertaken in children with HB as collaboration among the SIOPEL, COG, and JPLT study groups [12,14].

The same classification criteria are used to evaluate the post-treatment extent of disease (POSTTEXT) of HB after CHT. While the PRETEXT staging is strictly related to the prognosis, the POSTTEXT is the most important factor in planning the surgical resection [11,12,13].

In 2013, the Children’s Hepatic Tumors International Collaboration (CHIC) created a shared international database to provide novel prognostic factors with the overarching goal of developing a common risk stratification scheme for HB [3]. The CHIC database includes data from 1605 children treated in eight multicentre HB trials based on cisplatin–CHT and complete surgical resection over 25 years. The CHIC study analyzed the PRETEXT staging by centrally reviewing the tumour imaging (CT or magnetic resonance) [3]. The initial CHIC univariate analysis identified several factors associated with poor outcomes in children with HB: PRETEXT IV, macrovascular hepatic or portal involvement, contiguous extrahepatic disease, multifocality of the primary tumor, metastatic disease, tumor rupture, low alpha–fetoprotein (AFP) levels (<100 ng/mL) or high AFP levels (>1 million ng/mL) at diagnosis, age (≥8 years), low birth weight (<1500 gr), prematurity, and comorbidities (Beckwith-Wiedemann syndrome) [3].

Based on these specific prognostic factors, the CHIC group recently developed a new HB risk-stratified staging system, named CHIC–HB Stratification (CHIC–HS), according to five group classifications which showed a 5-year EFS of 86% for PRETEXT I/II, 82% for PRETEXT III, 60% for PRETEXT IV, 42% for metastatic disease, and 35% for AFP concentration ≤ 100 ng/mL at diagnosis [2]. Age at diagnosis, AFP levels, and PRETEXT stage remained statistically significant prognostic factors for all sub-groups [2]. Based on those variables, each patient is assigned to a risk group, which defines his prognosis as very low/low risk (EFS ≥ 89%), intermediate risk (EFS of 50–88%) or high risk (EFS < 50%) [2]. Currently, the CHIC–HB stratification is a useful tool to define the prognosis of children diagnosed with HB, and it is being currently validated in the PHITT trial [14]. Moreover, in future prospective the new CHIC risk stratification might potentially also direct the therapeutic management of HB.

In the CHIC–HS, the presence of metastatic disease at diagnosis is defined as “high-risk”, independent of the PRETEXT staging, and has been found in 20% of the study population. For the sub-group of children affected by metastatic HB, the AFP level was identified as the most significant prognostic factor: an AFP level of 100–1000 ng/mL at diagnosis was related to a 5-year EFS of 18%, while children with AFP > 1000 ng/mL had a 5-year EFS of 47% (*p* < 0.0001) [2]. The second most significant prognostic factor for metastatic HB was age at presentation, with worse prognosis for children aged ≥ 8 years. Lastly, PRETEXT III–IV and the presence of one or more of PRETEXT annotation factors (such as involvement of IVC, portal vein, extrahepatic contiguous tumour extension, multifocal liver lesions, or tumor rupture at diagnosis) also were associated with inferior outcomes [2,3].

## 4. Hepatoblastoma (HB) Treatment Algorithms

HB treatment requires a multi-modal approach of CHT combined with surgery, and it strictly relies on the PRETEXT and POSTTEXT staging of the tumour, its biology, presence of metastases, and response to neoadjuvant CHT (primary tumour mass reduction, decrease in AFP levels). The goal of HB therapy is radical extirpation of the tumor. There have been substantial improvements in HB treatment in recent years: from a few treatment options, there is now a detailed tumor risk-management plan that includes CHT and several surgical approaches [2,4,15].

According to the principal protocols [5,6,15,16,17,18,19,20], at diagnosis the assessment of HB includes: (1) total body CT scan to define the PRETEXT stage with the annotation factors (V,P,E,F,R); (2) tumor biopsy in order to determinate the HB histological subtype (well-differentiated fetal HB, embryonal HB, small cell undifferentiated HB), which have been associated to different prognosis [13]; and (3) AFP blood levels. Based on those results, each trial followed a treatment algorithm, which may differ for the role of surgical resection at diagnosis and CHT regimens, and have been extensively described [6,15,16,17,18,19,20]. As first line-therapy, children may be treated by upfront surgery or by neoadjuvant CHT with subsequent surgical resection of the primary tumor. In COG guidelines, upfront primary tumor resection at diagnosis is recommended only in selected patients with PRETEXT I and II tumors (“very low” risk) with absence of macrovascular involvement defined as more than 1 cm radiographic free tumor margin from the middle hepatic vein, the retrohepatic inferior vena cava, and the portal bifurcation [2,5].

For PRETEXT III–IV or HB with synchronous metastatic disease (“high-risk”), all trials recommend neoadjuvant cisplatin-based CHT before considering any surgical resection. After 2–4 cycles of CHT, tumours should be re-imaged in order to outline the POSTTEXT stage with definition of surgical resectability of the primary liver tumor, the response of pulmonary metastases (if present at diagnosis) and whether further CHT is needed [13]. At this time point, focus on extra-hepatic tumor extension is pivotal in orienting the subsequent strategies (Figure 2).

Currently, the PHITT [14], which is a randomized interventional study based on the risk-adapted therapeutic approach, is exploring: (a) the reduction of chemotherapic treatment for low risk HB patients (aiming to maintain their excellent event free survival, but also to decrease acute and long-term toxicity); (b) intensification of therapy with the use of novel agents in the high-risk group; and (c) comparison of three different regimens in intermediate risk HB. A key strand of the PHITT trial is the evaluation of the biology of HB, using the identification/validation of novel and already reported prognostic biomarkers as well as toxicity biomarkers. In addition, the PHITT aims also to evaluate a surgical planning instrument for an impact on the decision-making process in POSTTEXT III and IV HB. Therefore, in the next future the PHITT results will provide practical tools to achieve personalized treatment in children with HB, not only for the CHT regimen but also for the surgical approach.

As the objective of our review, in the following paragraphs we summarize the management and outcomes of children affected by HB with synchronous lung metastasis reported in literature, focusing on the surgical treatment of the primary liver tumor (surgical resection, LT, non-surgical treatments) and lung metastases. 

## 5. Hepatoblastoma (HB) with Synchronous Lung Metastasis

Around 20% of children with HB have synchronous lung metastasis at diagnosis and these patients are burdened by an overall 25–50% lower survival range compared to those without [6]. As reported by the principal HB trials (Table 1), the outcomes of children with HB and synchronous lung metastases have significantly improved in the last decades due to the intensification of CHT regimens, which permitted to increase the metastatic disease response to CHT from 31% to 97% of to CHT. So far, the overall survival rate of children with HB and metastases at diagnosis have been reported as 44–79% at 3 years and 27–57% at 5 years of follow up, while the EFS is 21–77% at 3-years and 21–28% at 5 years of follow up [5,6,15,16,17,18,19,20].

Across different protocols, children with HB and lung metastasis at diagnosis are treated with neoadjuvant CHT followed by surgical resection of the primary tumor +/− pulmonary metastases, according to the primary tumor resectability and lung metastasis response to CHT [5,6,15,16,17,18,19,20].

Following the selection criteria described above, to the best of our knowledge we identified 434 patients affected by HB with synchronous lung metastases described in literature between 1990 and 2018. All children underwent neoadjuvant CHT before any surgical treatment and 43.5% of children completely cleared lung metastases after neoadjuvant CHT. Primary tumour surgery was performed in 407 (93.8%) of children and the surgical treatment consisted of liver resection in 183 (45%) of cases, LT in 61 (15%) of cases, while in 162 (40%) of patients’ surgical details were not specified. Of 91 (21%) of patients undergoing surgical resection of lung metastases, 25 (27.5%) had lung metastasectomy after primary tumour surgery, 22 (24.2%) before (mainly for LT) and 14 (15.4%) simultaneous to primary tumour surgery. After a median follow up of 4 years, the overall survival rate was 83% and recurrence rate 40% (Table 2).

### 5.1. Resectable Primary Liver Tumor and Synchronous Lung Metastasis

For children with HB and synchronous lung metastases who present resectable primary liver tumor (according to the POSTEXT staging) after neoadjuvant CHT, surgical treatment of the primary tumor follows the recommendations defined for HB without pulmonary metastases, by resecting the hepatic segments involved within the tumor [4]. However, the management of lung disease not cleared by neoadjuvant CHT varies greatly in terms of surgical approach and timings and not universal recommendation have been defined yet.

In 2007, Meyers et al. [17] reported the first results from the COG (INT–0098 protocol). They described 9 of 38 children affected by pulmonary metastatic HB at diagnosis, who underwent metastasectomy for initial pulmonary metastasis after neo-adjuvant CHT at varying points respect the surgery of the primary tumor (2 before, 5 simultaneous, and 2 after liver resection). Of those, 8 (88.9%) survived at long-term follow-up and 3 had a tumor relapse. From those results, the authors suggest that the management of metastases present at diagnosis and persisting after neoadjuvant CHT should be more aggressive—and lung metastasectomy should be strongly considered—providing the primary liver tumour is under control.

In 2013, Zsiros et al. [6] reported the excellent results of SIOPEL–4, characterized by increased dose of cisplatin in the neoadjuvant CHT. Compared to the older SIOPEL studies [18,22,23], the most important progress from SIOPEL-4 was demonstrated in the sub-group of children with HB and synchronous lung metastasis: out of 39 patients, 97% responded to neoadjuvant CHT (20 complete responses, 18 partial responses, 1 had no evaluation of lung lesions), with a 3-year EFS of 77% (95% CI 63–90) and 3-year survival of 79% (95% CI 66–92) [6]. Complete tumor primary resection was achieved in 70% of cases. Surgical treatment of the primary liver lesion in metastatic HB included liver resection for all patients, except for 7 children with unresectable HB who underwent LT after pulmonary metastases were cleared (6 with CHT, 1 with metastasectomy). Nineteen of the 20 HB metastatic patients with complete response in the lungs were in remission at the last evaluation, while in 1 patient, lung lesions recurred. Of the 18 patients with lung metastasis with partial response, after the primary tumor surgery, 7 children underwent metastasectomy (6 complete, 1 incomplete but nonviable tumor), and of these 4 children had complete remission. Six patients had residual lesions in the lung after the preoperative CHT but did not undergo pulmonary metastasectomy and achieved complete remission with further CHT. In 2 cases, lung lesions remained unresectable, and in one child, liver resection was performed without metastasectomy. In 1 patient, surgery was not attempted. In this series, only 1 (5%) patient had lung relapse at 3-year follow-up, results that demonstrate the stability of lung response achieved by the SIOPEL–4 CHT regimen. The SIOPEL–4 results emphasize that increased cisplatin dose in neoadjuvant CHT provides improved outcomes in metastatic HB, while metastasectomy remains the only curative option in patients whose lung lesions are not cleared by CHT alone.

In 2017, Hishiki et al. [16] reviewed the JPLT–2 prospective cohort study in order to clarify the role of pulmonary metastasectomy in HB with lung metastasis at diagnosis. Out of 60 patients affected by HB and synchronous lung metastases who underwent neoadjuvant CHT, 1 patient died from early CHT-related toxicity, while 59 completed the preoperative CHT scheme. Twenty-six (43%) children had a complete response to CHT, while in the other 33 with residual metastases, 14 underwent at least one metastasectomy before (*n* = 2) and after (*n* = 12) liver surgery. Among those who underwent metastasectomy, pulmonary nodules were completely removed from 11 children, whereas 3 had incomplete resection. In the same study, the metastasectomy group registered promising survival rate when considering EFS (45%) and overall survival (63.6%) at 3 years. Moreover, in children who underwent lung resection and liver surgery, the 3-year EFS increased to 55.6%, while it was 36.9% in patients who only experienced liver surgery. In this series, metastasectomy was restricted to cases that were not eradicated by CHT. Whereas the rate of clearance of initial HB metastasis in the JPLT–2 trial was slightly lower compared to those reported in the SIOPLE–4 (43% versus 51%, respectively), the authors state that intensifying CHT induction with the aim to control the lung disease, as in the SIOPEL–4 protocol, appears to be a reasonable strategy for improving the survival of patients with lung metastases. Consistent with the previous report from the same group, the JPLT–2 stated that among children with HB with lung metastasis at diagnosis, the best outcomes are achieved by complete response by CHT only, as expected. Yet, for cases that have residual disease after CHT, aggressive pulmonary metastasectomy may improve their outcome, provided that the liver tumor has been, or is expected to be, completely resected. Unfortunately, none of the abovementioned studies report on eventual surgical complications after metastasis resection.

Based on previous trials, in the current PHITT, children with HB and synchronous lung metastasis (high-risk group), after receiving initial 3 blocks of cisplatin-intensive SIOPEL–4 regimen, are stratified in 2 risk groups: patients with lung complete response will undergo resection of the primary tumor at any time after the completion of the induction therapy, followed by CHT consolidation; on the other hand, patients with lung metastases not cleared at the end of CHT induction are randomized to two different intensified consolidation therapies and surgical resection of the primary tumor can be considered at any time after the induction therapy, thus lung metastasectomy might be considered in all patients if continuing to respond to consolidation therapy [14].

### 5.2. Unresectable Primary Liver Tumor and Synchronous Lung Metastasis

Children with HB and metastatic disease at diagnosis, whose primary tumor remains unresectable after neoadjuvant CHT (defined as multifocal HB involving all four liver sections, tumor located next to both the main portal vessels, at the hilum of the liver or next to all three hepatic veins) are candidates for LT only if the metastatic disease is cleared with chemotherapeutic regimens and/or with pulmonary metastasectomy [13]. Unresponsive or progressive metastatic disease after neoadjuvant CHT is a contraindication for LT, because even if the nodules can be surgically resected, microscopic foci of a chemoresistant tumor are highly probable [24,25]. Therefore, lung metastases in patients who appear to respond to CHT, but who do not entirely clear, might benefit from surgical metastasectomy to allow subsequent LT [26,27].

In SIOPEL–4 [6], out of 7 children with HB and lung metastasis at diagnosis who underwent LT, after they cleared pulmonary metastasis (6 with CHT, 1 with metastasectomy [no viable tumor cells found in the specimen]), none had pulmonary relapse at 3-year follow-up. Based on these promising results, the SIOPEL group emphasises that the presence of lung metastasis at diagnosis is not a contraindication for LT, provided that effective preoperative CHT is provided and lung lesions respond to treatment and are completely cleared before transplantation by CHT and/or metastasectomy. Recently, Triana Junco et al. [1] also reported excellent LT outcomes for unresectable HB with synchronous lung metastasis eradicated before transplantation (1-year survival: 93.3% ± 4.6%; 5-year survival: 86.4% ± 6.3%). This data demonstrated that the presence of metastasis at diagnosis (12.9% of the study population), if resolved before LT, was not a risk factor of tumour recurrence and did not influence long-term survivals.

From our research, we found 61 children with HB and synchronous lung metastasis at diagnosis undergoing LT: all patients received neoadjuvant CHT and in 39 (63.9%) cases lung metastases were cleared by CHT alone before LT, while in 22 (36%) cases lung lesions resistant to CHT were resected before LT.

Due to the high response of lung metastases to neoadjuvant CHT and optimal LT outcomes, several trials recommend that intensification of the chemotherapeutic regimens that aim to completely clear pulmonary lesions is needed, especially for children with unresectable primary tumors who may benefit from transplantation [3,5,6].

A large series from the Surveillance, Epidemiology, and End Results (SEER) database reported that outcomes of children with HB who underwent LT (for unresectable liver tumor) were similar to these achieved in patients who underwent liver resection (for resectable liver tumor), with a 5-year survival of 86.5% and 85.6%, respectively (hazards ratio (HR) = 0.58, 95% confidence interval (CI) 0.07–5.10) [13,28]. In the initial experience, primary LT for unresectable HB demonstrated much better results (85%) than salvage LT (40%) after a 10-year follow-up [1,29]. Nevertheless, recent evidence from the nationwide survey of the outcomes of liver donor liver transplantation (LDLT) in Japan [30] showed similar results for rescue LT and primary LT: Sakamoto et al. reported that children (*n*= 15) who received initial liver resection, followed by rescue LDLT, showed comparable outcomes compared to patients (*n*= 24) treated by primary LDLT (recurrence free-survival rate at 1 and 3 years after LDLT were 86.7% vs. 70.8% and 78% vs. 62.2%, respectively (*p* = 0.24)). These promising results are probably related to the improving of JPLT chemotherapy protocols and need to be confirmed in other series. Due to the limited organ availability and the fact that transplantation is not without morbidity and mortality, authors suggest that liver resection should be careful considered on a case-by-case basis in order to avoid LDLT for a patient who may have the chance to be cured by liver resection.

After LT, tumor relapse has been related to risk factors such as PRETEXT IV lesions, older age at diagnosis, and longer waiting-list time. Consequently, several studies suggest that the liver transplant surgical team should be involved in the management of children with HB from the early stages to optimize outcomes, as also adopted in the PHITT [14,31]. To reduce the waiting-list time, from 2010 in the Pediatric End-Stage Liver Disease (PELD) allocation system, additional “exception” points are assigned to children with HB in order prioritize them on the LT waiting list. In this context, the rapid availability of organs from deceased donors (due to the prioritization of HB children on the waiting list), as well as the increasing living-donor LT activity, play a key role in enhancing the survival rate [1].

Yet, in advanced HB with vascular involvement, the optimal treatment between extreme liver resection and LT is still under debate. Indeed, each option has crucial drawbacks: non-conventional liver resection with vascular reconstruction could expose the patient to tumor residual (R1) or to major complications [32], while LT is associated with long-term immunosuppression side effects (opportunistic infection, relapse or secondary neoplasm, transplant-related vascular and biliary complications [33]). Therefore, in children with lung metastasis resistant to CHT (not candidate to LT) and/or in countries where organ availability is limited, extreme liver resection should be considered on case-by-case basis [30].

However, the optimal long-term outcomes of children who undergo LT at these stages suggest that transplantation is paramount when associated with effective preoperative CHT, and it is increasingly indicated when HB includes major venous invasion and multifocal PRETEXT IV, along with the absence of metastatic disease [1].

### 5.3. Unresectable Primary Liver Tumor and Synchronous Lung Metastases Progressing during Neoadjuvant Chemotherapy

In patients whose tumors progress or remain unresectable after neoadjuvant CHT as well as in children who cannot undergo LT for residual lung metastases, interventional techniques (such as transarterial chemoembolization (TACE), high-intensity frequency ultrasound (HIFU), radiofrequency ablation (RFA)) have been increasingly used. The aims of these techniques include decreasing tumor progression, reducing tumor size in order to increase the feasibility of liver resection, and limiting systemic CHT toxicity [34]. Unfortunately, the experiences of non-surgical treatment in patients with HB and synchronous lung metastases are limited to a few cases, and the results focus on the local tumor response and do not report outcomes of lung metastases.

Based on adult experience, preoperative TACE has been the most widely used approach in paediatric HB, with the aim of reducing tumour volume (through providing intratumoral necrosis and decreasing intraoperative bleeding) and allowing complete hepatic resection. However, the reported cases in HB children are sparse, and the use of TACE followed by surgery does not always result in a complete resection. Nevertheless, different series confirm that the presence of lung metastases is associated to poor outcomes [35,36,37]. Li et al. [36] reported encouraging results of 12 children with advanced HB treated by TACE, which permitted a mean tumor reduction of 69% (of the initial volume) followed by primary tumour resection in 83% of cases, without significant TACE-related toxicity. Interestingly, one child with unresectable HB and pulmonary metastases first received CHT treatment, which cleared lung lesions, followed by TACE, which permitted a subsequent surgical resection. However, long-term outcomes are not reported.

HIFU is a technique that allows focal delivery of high-intensity ultrasound beams directly to the tumor to enhance cell death. In a controlled, non-randomized prospective trial, HIFU combined with TACE was used for the treatment of unresectable HB with metastatic disease, and outcomes were compared to those obtained in children treated only by neo-adjuvant CHT (cispltain–5 fluorouracil–vincristine) [34]. In this report, the combined HIFU–TACE treatment improved the rate of radical resection and provided a higher and more rapid tumor size reduction within 6 months compared to the CHT group, without major complications. From these results, the authors propose that the combination of HIFU–TACE can be considered as upfront treatment for local control of advanced HB, with the advantage of not increasing CHT drug resistance. The HIFU–TACE benefit could be especially relevant in patients with metastatic disease. However, these promising results should be confirmed by larger trials.

Yevich et al. recently reported that also RFA is a safe and effective reiterative therapeutic option (either for the treatment of the primary liver tumor or lung metastasis) in children with metastatic HB, whom surgical resection is clinically contraindicated [38]. Although stem cell transplantation has been explored in patients with unresectable HB as an upfront treatment, it has not yet been proven beneficial due to the risk of toxicity and lack of evident capacity of rending HB tumor resectable. Therefore, the use of stem cell transplantation in metastatic HB requires further investigation [39,40,41].

The effectiveness of new anticancer agents, such as sorafenib and regorafenib, are currently investigated and have been demonstrated to reduce tumor growth in HB cell lines in vitro [42,43]. To the best of our knowledge, in clinical setting the use of sorafenib, combined with CHT cisplatin-based, is limited to case reports with recurrent metastatic HB after LT as second-line therapy [44,45], while there are no clinical data regarding regorafenib in children with HB.

## 6. Surgical Lung Metastatectomy: When and How?

Due to the excellent response of the metastatic disease to CHT, all international trials recommend the surgical treatment of lung lesions only in the cases of remnant pulmonary metastasis after neoadjuvant CHT or pulmonary relapses [3,5,6,16]. Contraindications to metastasectomy include only the inability to achieve a complete resection while preserving adequate lung function and the presence of uncontrolled disease at the primary site [4]. Yet, the timing of lung metastasectomy (before, after, or simultaneous to the surgical resection of the primary tumor) as well as the optimal surgical approach to pulmonary metastasis is still under debate. To the best of our knowledge, a summary of all reports available in literature describing the surgical approach of the primary liver tumor and the management of lung metastasis in children with HB and synchronous pulmonary metastases is reported in Table 3. 

The AHEP0731 study results represent the largest prospective report that details the characteristics and outcomes of children with metastatic HB [5]. Among 29 children with HB and lung metastasis at diagnosis, 9 (31%) met Response Evaluation Criteria in Solid Tumor (RECIST) criteria (defined as nodules ≥ 10 mm), 20 (69%) patients had bilateral disease and 9 (31%) patients had unilateral disease. Ten (33%) underwent lung nodule metastasectomy after receiving CHT (median number of metastases removed: 3 [range 1–12]), with no reported complications, and 2 of them did not receive any additional CHT and remained alive with normal AFP levels. The overall 3-year EFS in patients with lung metastasis was 49%, while the overall survival rate was 62%. In this series, the presence of measurable disease by RECIST, the sum of nodule diameters greater than or equal to the cumulative cohort median size (22 mm, with a range from 2 to 209 mm), bilateral pulmonary disease, and ≥ 10 nodules were associated with an increased risk of lower EFS. Since lesions that fail to meet RECIST size criteria (< 10 mm) at diagnosis may contain viable tumor, whereas residual lesions at the end of therapy may constitute eradicated tumor/scar tissue, the radiological criteria alone seem to not be the optimal method for evaluating disease response and predicting the outcome in HB. The treatment decision for metastatic HB should consider the presence of lesions at imaging evaluation as well as the total nodule burden, change in size over time, lesion stability, and serum AFP levels. The authors suggest that lesion biopsy at the end of therapy should be entertained for patients in whom serum AFP levels fail to normalize and/or nodules fail to change or diminish significantly in size. Finally, the COG experience emphasises that lung metastasectomy may be beneficial in patients whose lung disease did not clear with CHT, especially to facilitate LT. Wanaguru et al. [62] also showed positive data, reporting the outcomes of 8 HB patients with lung metastatic disease at diagnosis who underwent the SIOPEL chemotherapeutic protocols and achieved an EFS of 62.5% and an overall survival of 100% after a 2-year follow up.

Most surgeons prefer to perform the pulmonary metastasectomy after the resection of the primary tumor, because the control of primary HB is associated with improved outcomes [6,16,75]. The drawback of this staged approach is that adjuvant CHT may be delayed, and this consequence may lead to a long time interval between pre- and post-operative CHT. Moreover, HB cells may synthetize hepatocyte growth factor, the levels of which appear to increase after surgical resection with subsequent stimulation of growth, invasion, motility, angiogenesis, and prolonged survival of the tumour cells [76]. Based on this concept, Urla et al. [64] proposed the simultaneous resection of primary liver tumour and lung metastasis as single-stage surgery. In this series, 7 children with metastatic HB at diagnosis had all lung metastasis removed at the time of the liver surgery (number of metastasis removed ranged from 1 to 8; in 3 cases, lung lesions were bilateral). Liver surgeries comprised 3 left-trisegmentectomy, 2 right-trisegmentectomy, 1 right hemihepatectomy with IVC reconstruction, and 1 left-hemihepatectomy. At the 5-year follow up, a tumor recurred in 2 (28.6%) children, and the overall survival rate was 83%. Additionally, Fuchs et al. reported the positive experience of 2 children with persistent lung lesions after neoadjuvant CHT who underwent lung metastasectomy simultaneously with the primary tumor resection without increasing post-operative morbidity [68].

Alternatively, other authors prefer the pulmonary metastasectomy before liver resection to avoid the effects of growth stimulation and tumour cell proliferation of metastasis triggered by hepatic growth factors secreted after major liver surgery [77]. As detailed in Table 1, most of the experiences of lung metastasectomy performed before surgical resection of the primary tumor were reported for children candidates to LT, because post-operative immunosuppression can trigger the growth of extrahepatic lesions.

The optimal surgical approach to pulmonary metastasectomy is still controversial, especially for bilateral lung lesions, for which some authors prefer metachronous bilateral thoracotomies [75], while others favor sternotomy [64]. Due to the lack of data on surgical complications after lung metastasectomy in children with HB, there is no evidence as to the best approach for this issue. However, significant experience on the surgical treatment of pulmonary metastasis from other paediatric solid tumors shows that unilateral thoracotomy, bilateral thoracotomy, or median sternotomy are well tolerated in children [4]. Lung wedge resection is recommended (independent of the surgical approach) to provide adequate residual lung function. In a series of 43 children who underwent lung resection for pulmonary metastasis from solid tumor, Erginel et al. [66] demonstrated that non-anatomical wedge resection is an ideal technique for pulmonary metastasectomy due to the preservation of lung parenchyma and decreased blood loss. This procedure ensures a negative margin to avoid tumor recurrence (none of the patients had local relapse). There is no clear limit to the number of metastases that can be resected; thus, in the decision-strategy, the aim of each surgery is to achieve clear lungs [4,24].

For small lesions that are not palpable or are invisible, indocyanine green (ICG) navigation surgery, using a fluorescence imaging system, is useful for HB lung metastasectomy [78]. ICG allows detection of lesions as small as 0.062 mm in diameter that are located 5–10 mm from the surface, but it is associated with 10–20% false positives. However, since lesions lying in deeper layers may go undetected by ICG, intraoperative pathological analysis is still required to confirm negative surgical margins. To improve ICG sensitivity, a novel overlay fluorescence imaging system, using real-time navigation with an endoscope with the PINPOINT system, was recently proposed to achieve complete resection, but further data are needed [79].

CT-guided localization followed by video-assisted thoracoscopic surgery is also proposed for the metastasectomy of tiny pulmonary lesions (<5 mm). However, this technique may be associated with complications, including pneumothorax, lung hemorrhage, and air embolism [80].

Due to the limited number of cases and the lack of long-term outcomes for children who undergo pulmonary resection for synchronous lung metastases, it is not yet possible to provide recommendations on the timing of resection of lung metastases that are resistant to CHT or the best surgical approach. Nevertheless, the choice should be based on the tumor biology, resectability of the primary tumors and the surgeon’s experience.

## 7. Conclusions and Future Perspectives

In recent years, there have been substantial improvements in HB treatment: from a few treatment options, there is now a detailed risk-stratified tumor management plan that includes CHT and several surgical approaches. While more children who face this illness now have better hopes to be cured, attention should be focused on those patients who are diagnosed with a metastatic disease, where the therapeutic decision-making can be difficult, because it is still not supported by evidence due to the lack of data. In this scenario, the primary goal of the therapeutic management is to clear lung metastasis and resect the primary tumor. Currently, the treatment of HB with synchronous metastases requires a multi-modal approach of neoadjuvant CHT followed by surgery, and it strictly relies on the PRETEXT and POSTTEXT stage of the tumor, its biology, and the efficacy of neoadjuvant CHT.

In most cases, lung metastases have a good response to neoadjuvant CHT. Children who completely clear lung metastasis by CHT and undergo radical resection of the primary tumor have shown good results with a 3-year patient survival of about 80%.

Contrarily, the inability to clear lung metastases at the end of the CHT treatment is one of the most negative prognostic factors that greatly affects EFS and overall survival. Hence, in these children, every effort should be made in order to ensure clearance of lung disease, both by intensification of CHT and/or metastasectomy. If the primary tumor is resectable, lung lesions resistant to CHT might be surgical resected before, after or simultaneously to the surgery of the liver tumor. Due to the limited number of cases and the lack of long-term outcomes of children who undergo HB lung metastasectomy, it is not yet possible to provide recommendations on the optimal timing of lung metastases resection and the best surgical approach. Therefore, the choice should be based on the tumor biology, resectability of the primary tumor, and the multidisciplinary team experience.

Children with HB and metastatic disease at diagnosis, whose primary tumor remains unresectable after neoadjuvant CHT (multifocal HB involving all four liver sections; tumor located next to both the main portal vessels, at the hilum of the liver or next to all three hepatic veins) are LT candidates only if the metastatic disease is cleared and have been associated with excellent outcomes (5-year survival rate of up to 86%). Therefore, the aggressive management of lung metastases, either by intensification of CHT and/or metastasectomy, is crucial given the high risk of tumour recurrence in the lungs caused by the post-transplant immunosuppression.

In the presence of unresectable HB who cannot undergo LT for residual lung metastasis, alternative loco-regional treatment (such as TACE, HIFU, or RFA) might be considered in order to reduce tumor size and increase the feasibility of liver resection.

In children with HB and lung metastasis resistant to CHT, defining the viability of lung residual lesions is of paramount importance in order to choose the surgical approach for the primary liver tumor. So far, only resection and histological evaluation of the metastatic lesions allow clinicians to define the extra-hepatic extension of the disease. However, the surgical resection of small, diffuse metastatic lesions might be difficult to afford both by standard imaging and direct examination of the lungs. To overcome this limitation, novel techniques, such as ICG fluoroscopy, appear to be useful tools, but additional data are required to verify their utility.

In future prospective, due to the limited data available, more evidence from large-scale multicentre clinical trials focused on children with HB and synchronous lung metastasis are needed to define a tailored therapeutic management and to further improve outcomes.

## Figures and Tables

**Figure 1 cancers-11-01693-f001:**
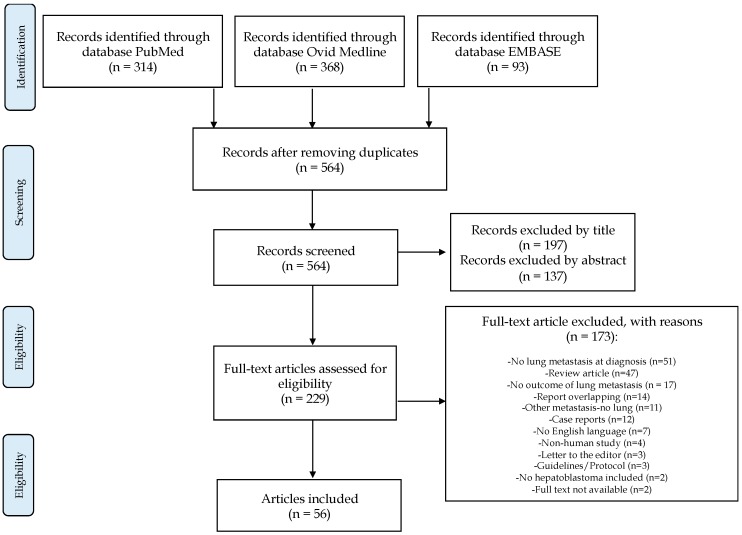
Flowchart of the literature search and study selection.

**Figure 2 cancers-11-01693-f002:**
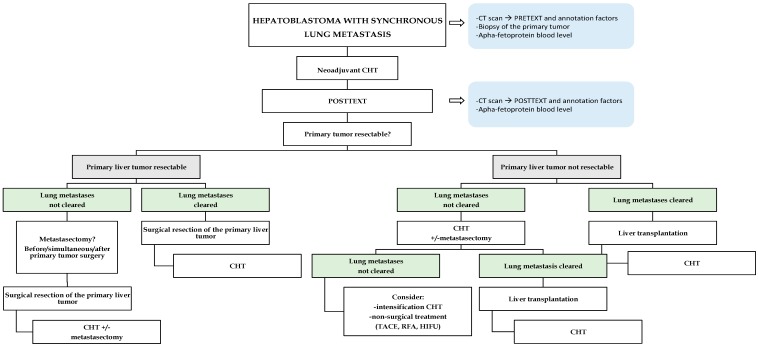
Decisional algorithm based on current protocols in children with hepatoblastoma and lung metastases at diagnosis. Abbreviations: CHT, chemotherapy; HIFU, high-intensity frequency ultrasound; LT, liver transplantation; PTETEXT, Pre-Treatment Extent of Disease; RFA, radiofrequency ablation; TACE, transarterial chemoembolization.

**Table 1 cancers-11-01693-t001:** Response and outcomes of different chemotherapeutic protocols in hepatoblastoma with distant metastases at diagnosis.

Response to CHT Protocols				Outcomes			
Study	*n* of Metastatic Patients	Non-Resected at Diagnosis (%)	Responders to Neoadjuvant CHT (%)	Eligible for Delayed Final Resection (%)	Resectable after Neoadjuvant CHT (%)	EFS (%)	Overall Survival (%)
COG (AHEP-0731) [5]	29	100	31	-	69	49 (3 years)	62 (3 years)
JPLT-2 [16]	35	100	43	-	-	21 (5 years)	44 (5 years)
SIOPEL-4 [6]	39	67.6	97	88.7	95	77 (3 years)	79 (3 years)
SIOPEL-3 [17]	70	98.7	71	83	91	56 (3 years)	62 (3 years)
SIOPEL-2 [18]	25	-	72	60	-	-	44 (3 years)
SIOPEL-1 [19]	31	-	84	-	-	28 (5 years)	57 (5 years)
INT-0098 [20]	40	98.4	-	57.1	47	25 (5 years)	37 (5 years)
POG-9345 [21]	11	-	-	36	-	27 (5 years)	27 (5 years)

Abbreviations: CHT: chemotherapy; EFS: event-free survival; JPLT, Japanese Study Group for Pediatric Liver Tumors; *n*, number; POG, Pediatric Oncology Group; SIOPEL, International Childhood Liver Tumour Strategy Group.

**Table 2 cancers-11-01693-t002:** Characteristics of patients affected by hepatoblastoma and synchronous lung metastases reported in literature.

Hepatoblastoma (HB) and Synchronous Lung Metastasis	Number (%, Range)
*n* of patients with HB and synchronous lung metastasis	434
*n* of patients with lung metastasis cleared after neoadjuvant CHT	189 (43.5%)
*n* of patients undergoing lung metastasectomy	91 (21%)
Timing of lung metastasectomy:	
-Before primary tumor surgery	22 (24.2%)
-After primary tumor surgery	25 (27.5%)
-Simultaneous to primary tumor surgery	14 (15.4%)
-Not specified	30 (32.9%)
Primary liver surgery:	
-Yes	407 (93.8%)
-No	27 (6.2%)
Type of primary tumor surgery:	
-Liver resection	183 (45%)
-LT	61 (15%)
-Not specified	163 (40%)
Median time to last follow up	4 (1–12)
Overall survival rate	83% (0–100%)
Overall recurrence rate	40% (0–100%)

Abbreviations: CHT, chemotherapy; HB, hepatoblastoma; LT, liver transplantation; *n*, number.

**Table 3 cancers-11-01693-t003:** Literature reports that describe the surgical management and outcomes for children with hepatoblastoma and synchronous lung metastasis.

Author	Year	*n* of Patients with HB and Synchronous Lung Metastases	Chemotherapy (CHT) Protocol	*n* of Patients Who Cleared Lung Metastases with Neoadjuvant CHT (%)	*n* of Patients with Synchronous Lung Metastases Undergoing Metastasectomy (%)	Timing of Lung Metastasectomy Respect the Primary Liver Surgery (after/before/simultaneous)	Type of Primary Liver Surgery (Resection/LT)	Follow up	Overall Survival	Recurrence Rate (%)
Langevin A [44]	1990	1	ADR + CDDP	100%	-	-	Liver resection	2 years	100%	0 (0%)
Iwafuchi M [45]	1991	2	ADR + CDDP	-	2	After liver surgery	Liver resection	1–3 years	100%	0 (0%)
Al-Qabandi W [46]	1999	3	SIOPEL 1–2	3 (100%)	-	-	3 LT	2 years	100%	-
Dower NA [47]	2000	2	Cisplatin-based	1 (50%)	1 (50%)	After liver surgery	1 LT, 1 liver resection	2–3 years	100%	50%
Nishimura SI [48]	2002	2	DOX/CDDP	1 (50%)	1 (50%)	After liver surgery	2 liver resection	6 years	100%	50%
Srinivasan P [49]	2002	1	PLADO	1 (100%)	-	-	1 LT	8 months	100%	100%
Fuchs J [50]	2002	7	CDDP/IFO/DOX–VP16/CARBO (GCLPT study HB 94)	1 (14.3%)	na	na	1 liver resection, 6 no primary tumor surgery	5 years	85.7%	na
Matsunaga T [8]	2003	20	CDDP + THP–ADR	11 (55%)	3 (15%)	After liver surgery	19 liver resection (11 complete, 8 incomplete)	3 years	51.9%	40%
Dicken BBJ [51]	2004	13	Cisplatin-based (not specified)	7 (53.8%)	-	-	13 liver resection	5 years	56.5%	87.5% (7 initially cleared by CHT)
Khan AS [52]	2006	1	PLADO	1 (100%)	-	-	1 LT	5 years	100%	100%
Meyers [17]	2007	38	COG INT–0098 protocol	20 (52.6%)	9 (45%)	2 before/5 simultaneous/2 afterliver surgery	na	6–12 years	88.8% (all initial metastasectomy)	34.2%
D’Antiga [53]	2007	1	SIOPEL (type not specified)	-	-	-	-	5 years	0%	Persistent lung metastasis after CHT
Suh M [54]	2008	1	COG (C5V + doxorubicin)	-	1 (100%)	Before liver surgery	1 LT	2 years	0%	100%
Faraj W [55]	2008	2	SIOPEL 1–4	2 (50%)	-	-	2 LT	1–10 years	50%	50%
Kosola S [56]	2010	2	SIOPEL–1	2 (100%)	1 (50%)	Before liver surgery	2 LT	11 years	50% (1 died after 15 years disease-free)	50%
Zsiros J [57]	2010	70	SIOPEL–3	36 (52.2%)	2 (2.9%)	na	26 liver resection, 5 LT	4.5 years	62%	44%
Latuz TB [58]	2010	3	COG protocol (AHEP0731)	1 (33.3%)	2 (66.7%)	na	3 liver resection	5 years	100%	33.3%
Koh KN [59]	2011	16	2–cistplatin based (CCG–823F trial)	na	na	na	na	5 years	42.1 ± 12.8 %	na
Hery G [60]	2011	2	SIOPEL 3–4	2 (100%)	-	-	2 LT	4.4/4.8 years	100%	-
Ismail H [24]	2011	5	PLADO	2 (40%)	-	-	2 LT (3 unresectable)	3 years	50%	20% (after LT)
Cruz JR [61]	2013	8	Cisplatin-based (not specified)	6 (75%)	2 (25%)	Before liver surgery	2 LT	5 years	85%	25%
Zsiros J [6]	2013	39	SIOPEL 4	25 (64.1%)	7 (17.9%)	na	7 LT na liver resection	3 years	79%	5%
Wanaguru D [62]	2013	8	SIOPEL	5 (62.5%)	-	-	8 liver resection	2 years	100%	25%
Sakamoto S [30]	2014	3	JPLT–2	2 (66.7%)	1 (33.3%)	Before liver surgery	3 LT	2–4 years	100%	33.3%
Zhang Y [63]	2014	12	AEP/ACP protocols (APBSCT program)	-	7 (58.3%)	na	na	2 years	66.7%	na
Pham TA [33]	2015	7	COG protocols (Cisplatin-based)	na	na	na	7 LT	10 years	85.7%	28.6%
Urla C [64]	2015	7	na	-	7 (100%)	Simultaneous to liver surgery	7 liver resection	5 years	83%	28.6%
Samuk [65]	2016	2	na	1 (50%)	1 (50%)	Before liver surgery	2 LT	1 year	50%	50%
Erginel B [66]	2016	2	na	na	2 (100%)	na	na	3 years	50%	-
Shanmugam N [67]	2017	3	PLADO	3 (100%)	-	-	3 Liver resection	3 years	66%	33.3%
Hishiki T [16]	2017	60	JPLT–2	26 (43%)	14 (23.3%)	2 before/12 after liver surgery	2 LT 57 liver resection	3 years	63.6% for lung complete resection; 41.8% for lung tumor not incompletely resected	37.3%
Fuchs J [68]	2017	9	SIOPEL or GPOH (not specified)	7 (77.8%)	2 (22.2%)	simultaneous to liver surgery	Liver resection	5 years	88%	11.1%
Busweiler et al. [69]	2017	27	SIOPEL 1–4	na	7 (25.9%)	3 before/4 after liver surgery	5 liver resection 2 LT	5 years	na	22.2%
O’Neill A [5]	2017	29	COG protocol (AHEP0731)	10 (34.5%)	10 (34.5%)	na	1 LT 19 liver resection (9 not resected)	3 years	62%	44.8%
Khan AS [52]	2017	4	SIOPEL protocols	4 (100%)	-	-	4 LT	5 years	50%	50%
Isono [70]	2018	2	JPLT 1/2	1 (50%)	1 (50%)	Before liver surgery	2 LT	5 years	100%	50%
Dall’Igna [71]	2018	4	SIOPEL 2–3	na	na	na	na	5 years	75%	25%
Fonseca A [72]	2018	2	SuperPLADO/SIOPEL–4	2 (100%)	-	-	2 Liver resection (extended-right hepatectomy, 1 with vascular reconstruction)	2.9–1.7 years	100%	50%
Ramos-Gonzalez G [31]	2018	3	C5V	-	3 (100%)	Before liver surgery	3 LT	4 years	66.6%	30%
Uchida H [73]	2018	8	JPLT–2	4 (50%)	4 (50%)	Before liver surgery	3 LT 5 liver resection	2–5 years	100%	50%
Umeda K [74]	2018	3	ADR, CDDP, CBDCA	2 (66.7%)	1 (33.3%)	Before liver surgery	3 LT	5 years	66.6%	66.7%

Abbreviations: AD, adriamycin; CHT, chemotherapy; C5V, cisplatin/5-flurouracil/vincristine; CBCDA, carboplatin; CDDP, Cis–Dichloro–Diamine–Platinum; COG, Children’s Oncology Group; DOX, doxorubicin; EFS, event-free survival; JPLT, Japanese Study Group for Pediatric Liver Tumors; LT, liver transplantation *n*, number; SIOPEL, International Childhood Liver Tumor Strategy Group.

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
