# Peer review of "How Do Synchronous Lung Metastases Influence the Surgical Management of Children with Hepatoblastoma? An Update and Systematic Review of the Literature"

_cancers, 2019, doi:10.3390/cancers11111693_

Round 1

Reviewer 1 Report

Authors have made great comprehensive review as to surgical management for HB with synchronous lung metastasis. This certainly will help surgeons as well as oncologists plan treatment algorithm.

I have a few comments, to which I would like authors to respond.

1. Regarding the outcome of whether primary LT is better than rescue LT, I would like authors to mention outcome of rescue LT compares favorably to primary LT more recently, which was described by Sakamoto, et al(reference No 66).

This result could be explained by the recent progress in chemo-regimen. 

Since some surgeons prefer aggressive surgical liver resection especially in the presence of lung metastasis, every efforts should be made to avoid LT given the limited availability of organs.

2. Is there any new papers or data regarding new chemo regimen including sorafenib or regorafenib for high -risk patients with lung metastases? If any, please cite them.

3. In page 8, I think 3-year-EFS rate of children with HB+lung meta is 21-77(not 21-49%) is this correct?

4. In figrue 2, there are unreadable characters in blue squares. My Mac can be blamed for this issue though.

5.In page 16, what cm is the cumulative cohort median size of lung mets? Please describe it.

Author Response

Reviewer 1

Authors have made great comprehensive review as to surgical management for HB with synchronous lung metastasis. This certainly will help surgeons as well as oncologists plan treatment algorithm.

I have a few comments, to which I would like authors to respond.

1. Regarding the outcome of whether primary LT is better than rescue LT, I would like authors to mention outcome of rescue LT compares favourably to primary LT more recently, which was described by Sakamoto, et al(reference No 66).

This result could be explained by the recent progress in chemo-regimen. 

Since some surgeons prefer aggressive surgical liver resection especially in the presence of lung metastasis, every efforts should be made to avoid LT given the limited availability of organs.

Author response:

Thank your for your comments. As suggested, the National Survey of the outcomes of living donor liver transplantation (LDLT) for hepatoblastoma (HB) in Japan, described by Sakamoto et al. in 2014 (in the revisited manuscript, reference n. 30) underlined new perspectives regarding the role of recue LT, which showed similar outcomes of primary LT. In the Japanese experience, after transplantation children (n= 15, 38.5%) who received initial liver resection, followed by rescue LDLT, showed similar outcomes compared to patients (n= 24.61.5%) treated by primary LDLT [recurrence free-survival rate at 1 and 3 years after LDLT were 86.7% vs 70.8% and 78% vs 62.2%, respectively (p=0.24)]. These promising results are probably related to the improving of JPLT chemotherapy protocols and should be considered before choosing primary LT or liver resection for cases with likely unresectable HB due to the limited organ resource. Since rescue and primary LT showed similar outcomes and transplantation is not without morbidity and mortality, authors suggest that liver resection should be careful considered on a case-by-case basis in order to avoid LDLT for a patient who may have the chance to be cured by liver resection.

We included in the revisited manuscript details on the results of Sakamoto et al. regarding the outcomes of primary and rescue LDLT. A comment on the limited availability of liver organs for LT, which needs to be considered for the surgical choice, was also included in the revisited text.

2. Is there any new papers or data regarding new chemo regimen including sorafenib or regorafenib for high -risk patients with lung metastases? If any, please cite them.

Author response:

Sorafenib and regorafenib are increasingly used as new anticancer agents for hepatocellular carcinoma [Wang W et al. Cancers (Basel). 2019 Oct 9;11(10)], while they have been rarely employed in HB yet.

Eicher C et al. [Br J Cancer. 2013 Feb 5;108(2):334-41] showed that in vitro sorafenib reduce tumor growth in hepatoblastoma (HB) cell lines and xenografts and the combination of sorafenib and cisplatin led to a remarkable decrease in tumoral cell viability. These results posed the bases on the concept that in the clinical setting combination of sorafenib with cisplatin could represent a promising treatment option for high-risk or recurrent HB. So far, in clinical practice the use of sorafenib in HB is limited to case reports with recurrent metastatic HB after LT as second line treatment [Marsh AM et al. Pediatr Blood Cancer. 2012 Nov;59(5):939-40; Shanmugam N et al. Pediatr Blood Cancer. 2017 Dec;64(12)].

To the best of our knowledge, there are no clinical data regarding the use of regorafenib in HB. Nevertheless, Indersie et al. [Hepatol Commun. 2017 Apr 6;1(2):168-183] demonstrated that antitumoral drugs acting on microRNA (such as regorafenib) inhibits HB growth in vitro and in vivo; therefore, this agent might represent possibly in future new options in the therapeutic armamentarium for HB.

A comment on the current experience and future perspective of these new antitumoral agents (sorafenib and regorafenib) has been added in the revisited manuscript. Thank you for your comment.

3. In page 8, I think 3-year-EFS rate of children with HB+lung meta is 21-77(not 21-49%) is this correct?

Author response:

Thank you for pointing out the mistake, which has been corrected accordingly (3-year-EFS rate of 21-77%).

4. In figure 2, there are unreadable characters in blue squares. My Mac can be blamed for this issue though.

Author response:

I double-checked the issue you mentioned with different laptops and with my Mac, but I did not find any anomaly in Figure 2. Please do not hesitate to point-out again if this issue further occurs.

5.In page 16, what cm is the cumulative cohort median size of lung mets? Please describe it.

Author response:

The median size of lung metastasis is 22 mm, with a range from 2 to 209 mm. We inserted this information in the text of the revisited manuscript. Thank you for your kind revision.

Reviewer 2 Report

The authors achieved to perform a very good review of the questions about patients with pulmonary metastatic hepatoblastoma and all the current questions actually non resolved and needed to be discussed

Author Response

Reviewer 2

The authors achieved to perform a very good review of the questions about patients with pulmonary metastatic hepatoblastoma and all the current questions actually non resolved and needed to be discussed.

Author response:

Thank you for your kind comments.

Reviewer 3 Report

The authors present a nice overview of an important issue that is being studied in the current international liver tumor trial - The Pediatric Hepatic International malignancy Therapeutic Trial (PHITT). The authors  review the literature and summarize previous findings.

Critique:

Although it is a nice review, the manuscript does not acknowledge the PHITT protocol which has been widely available since 2018 on both COG and SIOPEL websites. PHITT has been open since 2017 and has enrolled patients from all over the world. Figure 2 is the surgical guideline that was developed for patients with metastatic disease which has been shared in multiple venues. As such, the authors need to reference the trial in multiple aspects of the manuscript. I am sure this is an oversight but this has to be addressed.  

Author Response

Reviewer 3

The authors present a nice overview of an important issue that is being studied in the current international liver tumor trial-The Pediatric Hepatic International malignancy Therapeutic Trial (PHITT). The authors review the literature and summarize previous findings.

Critique:

Although it is a nice review, the manuscript does not acknowledge the PHITT protocol which has been widely available since 2018 on both COG and SIOPEL websites. PHITT has been open since 2017 and has enrolled patients from all over the world. Figure 2 is the surgical guideline that was developed for patients with metastatic disease which has been shared in multiple venues. As such, the authors need to reference the trial in multiple aspects of the manuscript. I am sure this is an oversight but this has to be addressed.

Author response:

Thank you for your comment. Due to the nature of the current systematic review, including only studies reporting outcomes on HB children with synchronous lung metastasis, in the initial version of the manuscript we had mentioned the Pediatric Hepatic International malignancy Therapeutic Trial (PHITT) only in reference n. 12 [Towbin AJ et al. Pediatr Radiol. 2018 Apr;48(4):536-554], reffering to the revisited PRETEXT radiologic staging system defined by the PHITT. Additionally, to avoid risk of redundancy [since we have largely referred in the main text to the results of the trial from the study groups included in the PHITT – International Society of Paediatric Oncology Epithelial Liver Tumor Group (SIOPEL), Liver Tumor Committee of the Children’s Oncology Groups (COG), the Japanese Children’s Cancer Group (JCCG)] and due to the limit of world count, we didn’t report further data about the PHITT trial, which is still on going.

As you kindly mentioned, PHITT is undoubtly the largest current single clinical trial undertaken in paediatric liver cancer patients as collaboration between the largest study groups on this topic (SIOPEL, COG and JCCG). The protocol is based on: 1) reducing treatment for low risk HB patients (aiming to maintain their excellent event free survival, but also to decrease acute and long-term toxicity); 2) intensification of therapy with the use of novel agents in the high-risk group: 3) comparing three different regimens in intermediate risk HB. Moreover, a key strand of this trial is the evaluation of the biology of HB, using the identification/validation of novel and already reported prognostic biomarkers as well as toxicity biomarkers. Therefore, in the next future the PHITT results will definitely provide new tools and therapeutic strategies for a personalized therapy in HB.

In the revisited manuscript, we described the PHITT trial as well as its endpoint and the future implications in the HB management in different sections of the text (Cap 3. HB Risk stratification; Cap. 4 HB treatment algorithms; Cap 5. HB with Synchronous lung metastasis).

Thank you for your comment. We believe that adding information on the PHITT provided important data about the future perspective in the therapeutic management of HB.

Round 2

Reviewer 3 Report

The author have addressed the previously raised concerns.